# Clinical outcomes and inflammatory marker levels in patients with Covid-19 and obesity at an inner-city safety net hospital

Anahita Mostaghim[1], Pranay Sinha[2], Catherine Bielick[1], Selby Knudsen[2], Indeevar Beeram[3], Laura F. White[4], Caroline Apovian[5], Manish Sagar[2], Natasha S. Hochberg[2,6]*

1 Department of Internal Medicine, School of Medicine, Boston University, Boston, Massachusetts, United States of America, 2 Section of Infectious Diseases, Department of Internal Medicine, School of Medicine, Boston University, Boston, Massachusetts, United States of America, 3 School of Medicine, Boston University, Boston, Massachusetts, United States of America, 4 Department of Biostatistics, Boston University School of Public Health, Boston, Massachusetts, United States of America, 5 Section of Endocrinology, Diabetes, Nutrition, and Weight Management, Department of Internal Medicine, School of Medicine, Boston University, Boston, Massachusetts, United States of America, 6 Boston Medical Center, Boston, Massachusetts, United States of America

* nhoch@bu.edu

**Data Availability Statement:** All relevant data are within the manuscript and its supporting information files.

## Abstract

### Objectives

Patients with Covid-19 and obesity have worse clinical outcomes which may be driven by increased inflammation. This study aimed to characterize the association between clinical outcomes in patients with obesity and inflammatory markers.

### Methods

We analyzed data for patients aged ≥18 years admitted with a positive SARS-CoV-2 PCR test. We used multivariate logistic regression to determine the association between BMI and intensive care unit (ICU) transfer and all-cause mortality. Inflammatory markers (C-reactive protein [CRP], lactate dehydrogenase [LDH], ferritin, and D-dimer) were compared between patients with and without obesity (body mass index [BMI] ≥30 kg/m$^2$).

### Results

Of 791 patients with Covid-19, 361 (45.6%) had obesity. In multivariate analyses, BMI ≥35 was associated with a higher odds of ICU transfer (adjusted odds ratio [aOR] 2.388 (95% confidence interval [CI]: 1.074–5.310) and hospital mortality (aOR = 4.3, 95% CI: 1.69–10.82). Compared to those with BMI<30, patients with obesity had lower ferritin (444 vs 637 ng/mL; p<0.001) and lower D-dimer (293 vs 350 mcg/mL; p = 0.009), non-significant differences in CRP (72.8 vs 84.1 mg/L, p = 0.099), and higher LDH (375 vs 340, p = 0.009) on the first hospital day.

### Conclusions

Patients with obesity were more likely to have poor outcomes even without increased inflammation.

**Funding:** Boston Obesity Nutrition Research Center P30DK046200 (CA), NIH K24 AI-145661 (MS), NIH 5T32AI152074-13 (PS), Warren Alpert Foundation (NSH).

**Competing interests:** The authors have declared that no competing interests exist.

## Introduction

Coronavirus disease 2019 (Covid-19), first reported in Wuhan, China, has since spread throughout the globe and been declared a pandemic by the World Health Organization [1]. It has an observed case fatality ratio ranging from 1.4% to 15.4%, with around 3.9% reported in the USA [2]. Mortality rates are higher in hospitalized patients and those with medical comorbidities [3–6] and reach 50% in those requiring the intensive care unit (ICU) [3]. Multiple cohort studies suggest those who are overweight or have obesity are more likely to experience invasive mechanical ventilation, ICU admission, or death [6–15]. Indeed, the OPEN-SAFELY study reports stepwise greater mortality with higher body mass index (BMI) strata [12]. Population-level studies also suggest a higher Covid-19 mortality rate in countries with greater prevalence of obesity [16].

Severe Covid-19 is hypothesized to be caused by cytokine release syndrome (CRS), an inflammatory immune response leading to organ failure [17, 18]. Severe Covid-19 and CRS have been linked to elevated levels of interleukin (IL)-6 [19–21] which stimulates the liver to produce C-reactive protein (CRP) and fibrinogen [22]. In addition to CRP and fibrinogen, lactate dehydrogenase (LDH) and ferritin correlate with plasma IL-6 levels [23, 24]. Serum LDH correlates with IL-18 production by activated macrophages [25] and elevated CD8+ cytotoxic T-cell activity in severe and chronic pulmonary infections [26]. Lymphocyte destruction or direct tissue damage from microorganisms, inflammation, or tissue ischemia may mediate the increased LDH [27]. Although multiple organs can be injured in the setting of Covid-19, the most prominent site is the lungs; patients can progress to acute respiratory distress syndrome (ARDS) and develop microthrombi or pulmonary emboli due to hypercoagulability [28, 29].

Though definitive reasons for poor Covid-19 outcomes in obesity remain uncertain, patients with obesity are uniquely vulnerable. Patients with obesity may have independent risk factors for poor outcomes in Covid-19 (type 2 diabetes (T2D), hypertension, and coronary artery disease [30]–conditions that are inflammatory and immune-mediated [31, 32]. Furthermore, obesity is associated with decreased functional residual capacity of the lungs. Fat deposition over the upper airway and thorax leads to difficulty with ventilation and peripheral lung collapse while in the supine position, which can cause hypoxemia [7, 33, 34]. Further, while having class III obesity (BMI $\geq$ 40) is associated with very poor clinical outcomes during critical illness, retrospective data shows a paradoxically protective association in the BMI group 30–34.9 (class I obesity) [7, 35, 36].

The observed increase in risk and severity may be associated with elevated inflammatory markers at baseline and a greater risk of CRS [7, 15, 34, 37]. Patients with obesity reportedly have chronic inflammation due to sustained production of proinflammatory cytokines in adipose tissue and higher levels of inflammatory cytokines, such as IL-6, CRP, and certain adipokines [34]. Additionally, within adipose tissue pro-inflammatory macrophage subsets (M1 phenotype) have been shown to replace the anti-inflammatory M2 phenotype macrophages [38].

In this case control study, we compared outcomes and inflammatory markers in patients with Covid-19 stratified by BMI and T2D to understand potential factors associated with the observed higher COVID-19 morbidity and mortality associated with obesity.

## Methods

### Study design, setting, participants

We conducted a retrospective cohort study at our medical center, a large safety-net hospital that primarily serves socio-economically disadvantaged patients with high rates of comorbid medical conditions [39]. We included patients aged $\geq$18 years who were hospitalized with a positive SARS-CoV-2 polymerase chain reaction (PCR) test between March 4 and May 1, 2020.

We obtained demographic and clinical information through electronic records query and manually abstracted inflammatory marker measurements (CRP, LDH, ferritin, and D-dimer), fraction of inspired oxygen (FiO2) requirements, and outcomes. Clinical outcomes assessed included ICU transfer and all-cause mortality. Patients discharged to hospice were classified as deceased. All activities associated with this project were approved by the Boston University Medical Center Institutional Review Board with waiver of informed consent to access non-anonymized patient data. Patient medical records from Boston University Medical Center were accessed from April to June 2020.

## Institutional procedure

Admission and daily laboratory order sets for patients with Covid-19 included standard testing (e.g., complete blood counts) and inflammatory markers (CRP, LDH, ferritin, and D-dimer). Supplemental oxygen for hospitalized Covid-19 patients excluded aerosolizing modalities such as high-flow nasal cannula or non-invasive positive pressure ventilation. Other procedural changes during this study's time period included encouragement to self-prone beginning on March 13, 2020.

All treatment decisions were at the discretion of the treating physician. Hydroxychloro-quine and azithromycin for a 5-day course were recommended for all Covid-19 patients admitted to the hospital with QTc<500ms, and on April 10[th] colchicine replaced azithromycin for non-pregnant patients without hepatic or renal disease. These medications were discontinued after receipt of a biologic or transfer to the ICU. The use of hydroxychloroquine, azithromycin, and colchicine was discontinued on April 23, 2020 after evaluation of newly published literature [40–42] and internal data review.

Treatment with "biologic therapy" included IL-6 receptor antagonists, such as tocilizumab or sarilumab, which was considered in patients with hypoxemia and elevated inflammatory markers (CRP>100mg/L or LDH >450 U/L). Anakinra was recommended for patients with hypoxemia and ferritin >5000 ng/mL. These biologic therapies were not recommended for patients with suspected or confirmed bacterial infection, severe heart failure, metastatic/stage IV cancer not in remission, severe ARDS, refractory shock, or SOFA score >11 [43].

## Statistical methods

We classified patients with BMI 25 to <30 kg/m$^2$ as overweight, and those with BMI$\geq$ 30 kg/m$^2$ as having obesity. We compared characteristics and outcomes between patients who had obesity and those who did not using fisher's exact test for categorical variables, logistic regression, Mann-Whitney U test for pairwise comparison of continuous variables and Kruskal-Wallis for comparison across T2D and obesity categories. Inflammatory markers were also compared across BMI strata: <25, 25 to 24.9, 30 to 34.9, 35 to <40, and $\geq$40 kg/m$^2$. For multivariate regression, we categorized obesity as BMI $\geq$30 compared to a reference of BMI<30. We included variables with p<0.2 in univariate analysis as covariates in the multivariate analysis and used backward elimination to build the model. We included T2D and a T2D-obesity interaction term. We considered two-sided p-value less than 0.05 statistically significant. We used SPSS v. 26.0 (IBM, Armonk, NY).

## Results

### Baseline demographics

A total of 791 patients were included. The median age was 65 years (interquartile range [IQR]:20) with 460 (58.2%) male and 363 (45.9%) with obesity (Table 1). The most common

**Table 1. Demographic characteristics and inflammatory markers on hospital days 1 and 2 among patients with COVID-19 with (BMI≥30) and without obesity (BMI<30), Boston, MA (n = 791).**

| Characteristic | Total (n = 791) | Obesity (n = 363) | Without Obesity (n = 428) | p-value |
|---|---|---|---|---|
| **Demographics** | | | | |
| Age (Median (IQR)) | 65 (20) | 57 (21) | 63 (24) | <0.001 |
| Male Sex, n (%) | 460 (58.2%) | 170 (47.1%) | 290 (67.4%) | <0.001 |
| **Comorbidities** | n (%) | n (%) | n (%) | |
| Diabetes | 223 (28.2%) | 105 (29.1%) | 118 (27.4%) | 0.634 |
| Hypertension | 348 (44.0%) | 166 (46.0%) | 182 (43.2%) | 0.168 |
| CAD | 56 (7.1%) | 21 (5.8%) | 35 (8.1%) | 0.214 |
| CHF | 20 (2.5%) | 7 (1.9%) | 13 (3.0%) | 0.371 |
| COPD | 38 (4.8%) | 17 (4.7%) | 21 (4.9%) | 1 |
| Asthma | 71 (9.0%) | 45 (12.5%) | 26 (6.0%) | 0.002 |
| CKD | 25 (3.2%) | 16 (4.4%) | 9 (2.1%) | 0.068 |
| HIV | 14 (1.8%) | 9 (2.5%) | 5 (1.2%) | 0.183 |
| Cancer | 6 (0.8%) | 3 (0.8%) | 3 (0.7%) | 1 |
| ESRD | 6 (0.8%) | 1 (0.3%) | 5 (1.2%) | 0.228 |
| **Day 1 labs** | Median (IQR) | Median (IQR) | Median (IQR) | |
| CRP (mg/L) | 76 (105) | 73 (81) | 84 (118) | 0.099 |
| LDH (U/L) | 354 (180) | 375 (176) | 340 (172) | 0.009 |
| Ferritin (ng/mL) | 541 (986) | 444 (661) | 637 (1114) | <0.001 |
| D-dimer (µg/mL) | 322 (180) | 293 (342) | 350 (495) | 0.009 |
| ALC (1000/µL) | 1.1 (0.8) | 1.1 (0.8) | 1.2 (0.8) | 0.001 |
| **Day 2 labs** | Median (IQR) | Median (IQR) | Median (IQR) | |
| CRP (mg/L) | 84 (100) | 98 (92) | 91 (114) | 0.329 |
| LDH (U/L) | 333 (188) | 317 (181) | 344 (188) | 0.005 |
| Ferritin (ng/mL) | 521 (936) | 410 (660) | 641 (1184) | <0.001 |
| D-dimer (µg/mL) | 332 (468) | 303 (382) | 373 (544) | 0.006 |
| ALC (1000/µL) | 1.2 (0.7) | 1.1 (0.7) | 1.3 (0.7) | 0.003 |
| **Supportive measures** | Median (IQR) | Median (IQR) | Median (IQR) | |
| FiO2 max | 30 (79) | 30 (76) | 27 (51.3) | 0.024 |
| IL-6 inhibitor use | 215 (27.2%) | 98 (22.9%) | 117 (32.4%) | 0.003 |

CAD = coronary artery disease, CHF = congestive heart failure, COPD = chronic obstructive pulmonary disease, CKD = chronic kidney disease, HIV = human immunodeficiency virus, ESRD = end-stage renal disease, CRP = C reactive protein, LDH = lactate dehydrogenase, ALC = absolute lymphocyte count, BMI = body mass index.

comorbidities were hypertension (n = 348, 44.0%) and T2D (n = 223, 28.2%). A total of 572 (72.3%) patients received supplemental oxygen and 244 (30.8%) patients received biologic therapy. The median time to biologic therapy administration was 1.2 hospital days (IQR: 1.15) after admission.

As compared to patients without obesity, those with obesity had a lower median age (57 vs. 63 years; p<0.001), were less likely to be male (47.1% vs. 67.4%; p<0.001), and were more likely to have asthma (12.5% vs. 6.0%; p = 0.002) (Table 1). Patients with obesity were also more likely to be treated with a biologic agent (35.5% vs 27.0%, p = 0.01). Additionally, patients with obesity were more likely to need higher levels of oxygen support with a median maximum FiO2 of 30 (IQR: 76) as compared to 27 (IQR: 51.3) for those without obesity (Table 1).

## Clinical outcomes

Outcome data were available for 789 (99.7%) patients. A total of 82 (10.4%) died during their hospital stay or were discharged to hospice; ICU level of care was required in 187 (23.6%) patients and 120 (15.2%) patients required mechanical ventilation.

Patients with BMI 30–34.9 did not have significantly different rates of unadjusted hospital mortality compared to those without obesity (11.4% vs 9.4%, p = 0.42). In multivariable analyses, BMI≥35 was associated with an increased risk of all-cause mortality (adjusted odds ratio [aOR] = 4.27, 95% confidence interval [95%CI]: 1.69–10.82) after adjusting for sex, maximum FiO2 requirements, IL-6 administration, and LDH; this effect was not seen for those with BMI 30–34.9 (Table 2).

Additionally, patients with obesity experienced a greater rate of ICU transfer (27.1% vs 20.7%, p = 0.04). In multivariable analysis, after adjusting for sex, maximum FiO2 requirements, IL-6 administration, and LDH, BMI 30–34.9 and BMI≥35 were associated with higher odds of ICU transfer compared to BMI <30 (aOR = 2.22, 95%CI: 1.06–4.61 and 2.39, 95%CI: 1.07–5.31, respectively). Interaction between T2D and obesity was not significant for ICU transfer (p = 0.76) or all-cause mortality (p = 0.22).

## Inflammatory markers

On hospital day 1, patients with obesity were more likely to have lower median values for ferritin (444 vs. 637 ng/mL, p<0.001) and D-dimer (293 vs. 350 mcg/mL DDU; p = 0.009; Table 1).

**Table 2. Univariate and multivariate analyses of comorbidities, inflammatory markers on day 1, maximum FiO2, and IL-6 inhibitor use in relationship to ICU transfer.**

| | ICU transfer | | | | |
|---|---|---|---|---|---|
| | No | Yes | Odds Ratio (95% CI) | p-value | Adjusted Odds Ratio (95% CI) |
| Age in years | | | 1.005 (0.995–1.015) | 0.351 | - |
| Male Sex | 331 (54.8%) | 129 (69.0%) | 1.834 (1.294–2.600) | 0.001 | 2.67 (1.38–5.18) |
| DM | 173 (28.6%) | 50 (26.7%) | 0.909 (0.629–1.315) | 0.613 | - |
| HTN | 273 (45.2%) | 75 (40.1%) | 0.812 (0.582–1.133) | 0.22 | - |
| CAD | 48 (7.9%) | 8 (4.3%) | 0.518 (0.240–1.115) | 0.103 | - |
| CHF | 13 (2.2%) | 7 (3.7%) | 1.768 (0.695–4.498) | 0.283 | - |
| COPD | 25 (4.1%) | 13 (7.0%) | 1.730 (0.867–3.454) | 0.12 | - |
| Asthma | 58 (9.6%) | 13 (7.0%) | 0.703 (0.376–1.314) | 0.307 | - |
| CKD | 19 (3.1%) | 6 (3.2%) | 1.021 (0.402–2.594) | >0.999 | - |
| HIV | 13 (2.2%) | 1 (0.5%) | 0.244 (0.032–1.881) | 0.207 | - |
| Cancer | 3 (0.5%) | 3 (1.6%) | 3.266 (0.654–16.321) | 0.148 | - |
| ESRD | 6 (1.0%) | 0 (0.0%) | 0.762 (0.733–0.792) | 0.345 | - |
| CRP | | | 1.006 (1.004–1.008) | <0.001 | - |
| LDH | | | 1.003 (1.001–1.004) | <0.001 | 1.00 (0.99–1.00) |
| Ferritin | | | 1.000 (1.000–1.000) | <0.001 | - |
| D-dimer | | | 1.001 (0.997–1.004) | 0.293 | - |
| ALC | | | 0.995 (0.982–1.010) | 0.524 | - |
| Max FiO2 | | | 1.062 (1.054–1.070) | <0.001 | 1.07 (1.06–1.09) |
| IL-6 inhibitor use | 124 (20.5%) | 91 (48.7%) | 3.669 (2.591–5.197) | <0.001 | 0.57 (0.30–1.10) |
| BMI <30 | 339 (56.6%) | 89 (47.6%) | | 0.097 | Reference group |
| BMI 30–34.9 | 132 (22.0%) | 50 (26.7%) | | | 2.22 (1.06–4.61) |
| BMI >35 | 128 (21.4%) | 48 (25.7%) | | | 2.39 (1.07–5.31) |

CAD = coronary artery disease, CHF = congestive heart failure, COPD = chronic obstructive pulmonary disease, CKD = chronic kidney disease, HIV = human immunodeficiency virus, ESRD = end-stage renal disease, CRP = C reactive protein, LDH = lactate dehydrogenase, ALC = absolute lymphocyte count, BMI = body mass index.

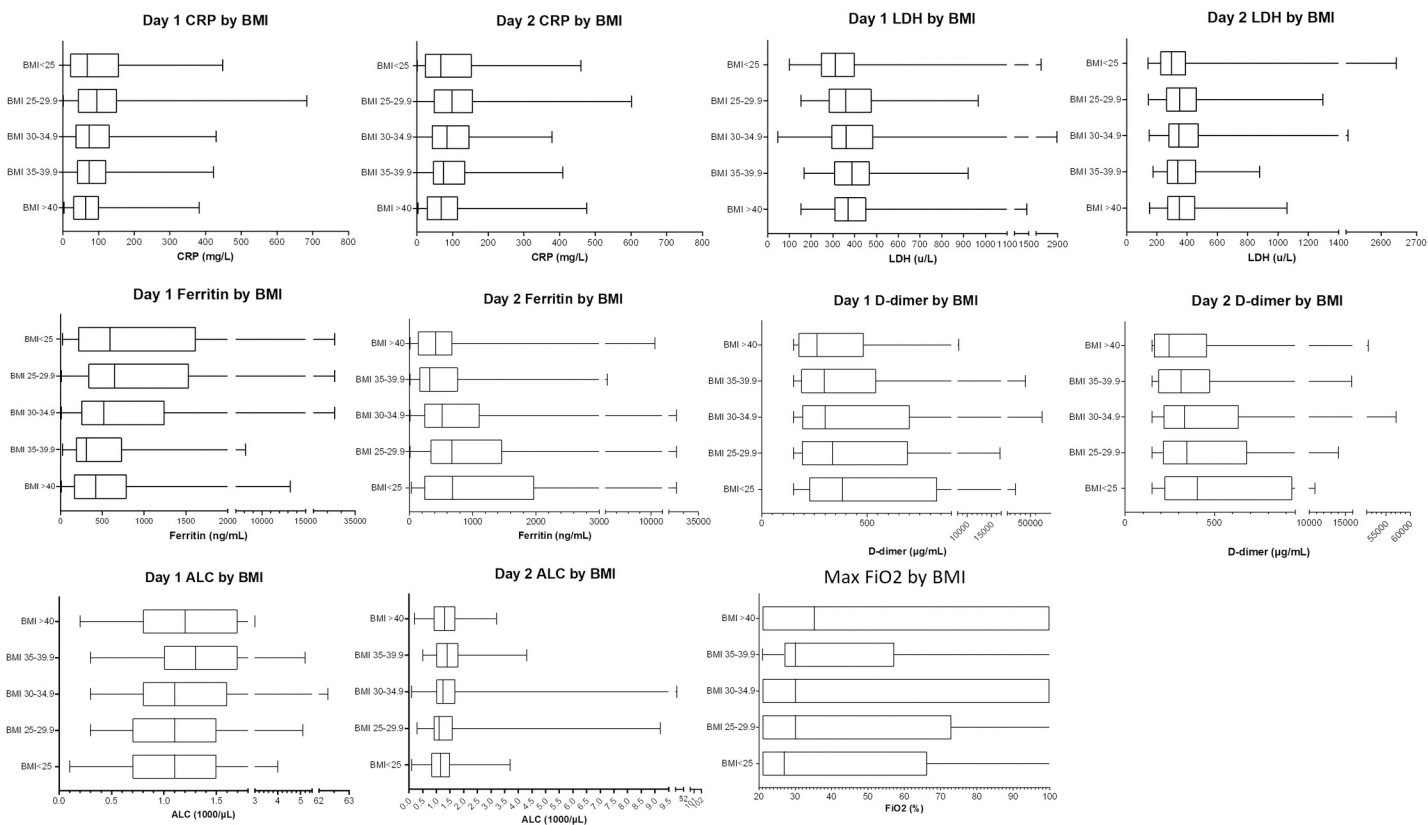

**Fig 1. Median, IQR, and range of inflammatory markers and absolute lymphocyte count on hospital days 1 and 2 as well as maximum FiO2 during hospitalization by body mass index (BMI) group.** CRP = C reactive protein, LDH = lactate dehydrogenase ALC = absolute lymphocyte count. * = 0.01 < p < 0.05, ** = p<0.01.

CRP values on day 1 were not significantly different compared to without obesity (73 vs 84 mg/L; p = 0.099). Patients with obesity had higher median values for LDH (375 vs. 340 U/L; p = 0.009) and absolute lymphocyte count (1.2 vs. 1.0 K/µL; p = 0.001). On day 2, median values of LDH, ferritin, and D-dimer were all lower in those with obesity compared to those without obesity (Table 1). There was no difference between median CRP in the two groups. When further stratified by BMI, median values of CRP and ferritin were highest in those with BMI 25–29.9, and median D-dimer was highest in patients with BMI<25. Median LDH was highest in patients with BMI 35–39.9 (Fig 1).

On day 1 of hospitalization, patients with both obesity and T2D had higher median values of CRP (98.7 vs. 83 mg/L, p = 0.0134), LDH (347 vs. 332 U/L, p = 0.038), and ferritin (660 vs. 531 ng/mL, p = 0.001) than patients with obesity who did not have T2D (Table 3). In patients without obesity, this pattern was not seen. In all patients with T2D compared to those without T2D, median CRP was significantly elevated (98.7 vs 83.6 mg/L, p = 0.01), but LDH and ferritin were reduced (332 vs. 344.5 U/L, p = 0.04 and 514 vs. 639 ng/mL, p = 0.0012, respectively).

By contrast, on day 2 of hospitalization, patients with both obesity and T2D had lower median LDH (302 vs. 328 U/L, p = 0.0058) and ferritin (567 vs. 725 ng/mL, p<0.0001), than those with obesity without T2D, but CRP was not significantly different. ALC across all four groups was significantly different between those with obesity with and without T2D on both day 1 and day 2 of hospitalization (Table 3).

**Table 3. Inflammatory markers and absolute lymphocyte count on hospital day 1 and maximum FiO2 required during hospitalization grouped by presence of obesity and diabetes.**

| | Both Obesity and Diabetes | Obesity only | Diabetes only | No diabetes and no obesity | p-value |
|---|---|---|---|---|---|
| | (N = 105) | (N = 256) | N = 118 | N = 312 | |
| Day 1 CRP (mg/L) | 98.7 (1.8–387.8) | 83 (0.4–683.3) | 81.4 (1.1–295.7) | 63.1 (0.3–427.5) | 0.013 |
| Day 2 CRP (mg/L) | 88.0 (1.0–417.2) | 88.5 (0.3–600.7) | 101.3 (1.4–358.5) | 68.6 (0.2–475.5) | 0.494 |
| Day 1 LDH (U/L) | 332 (154–720) | 347 (100–2824) | 391 (214–802) | 355 (47–2897) | 0.038 |
| Day 2 LDH (U/L) | 302 (141–811) | 328 (114–2643) | 362 (172–1014) | 337 (149–2507) | 0.006 |
| Day 1 Ferritin (ng/mL) | 531 (12–33,511) | 660 (26–33,511) | 494 (15–25,354) | 391 (9–33,511) | 0.001 |
| Day 2 Ferritin (ng/mL) | 567 (16–26,850) | 725 (13–33,511) | 504 (20–28,130) | 410 (12–33,511) | <0.001 |
| Day 1 D-dimer (μg/mL) | 382 (150–8,148) | 341 (111–44,213) | 295 (150–54,550) | 291 (150–48,048) | 0.075 |
| Day 2 D-dimer (μg/mL) | 398 (150–10,779) | 377 (150–14,027) | 303 (130–57,063) | 306.5 (150–51,299) | 0.054 |
| Day 1 ALC (1000/μL) | 1.1 (0.2–3.5) | 1.0 (0.1–5.1) | 1.2 (0.2–11.9) | 1.3 (0.2–62.3) | 0.002 |
| Day 2 ALC (1000/μL) | 1.2 (0.2–3.1) | 1.1 (0.1–9.2) | 1.3 (0.5–4.1) | 1.3 (0.1–51.2) | 0.015 |
| Max FiO2 | 27% (21–100) | 28.5 (21–100) | 34.5% (21–100) | 28.5 (21–100) | 0.013 |

CRP = C reactive protein, LDH = lactate dehydrogenase, ALC = absolute lymphocyte count, BMI = body mass index.

## Discussion

In this retrospective study, we compared outcomes and inflammatory markers in patients with and without obesity who were hospitalized with Covid-19 at a safety net hospital. Our data suggest that BMI ≥35 was associated with a two-fold increased risk of ICU transfer and a four-fold risk of all-cause mortality; BMI in the 30–34.9 range (Class I obesity) was also associated with increased risk of ICU transfer, but not significantly associated with increased mortality. We also found that Covid-19 patients with obesity had lower inflammatory markers on the first and second hospital days compared to those without obesity.

Our finding that patients with Class I obesity did not have increased Covid-19 mortality may reflect previously reported paradoxical outcomes in this patient group [42]. Prior reports of paradoxical outcomes were in previously unhealthy populations and believed secondary to an increased prevalence of catabolic state in those with normal range BMI [7]. In Covid-19, previously healthy patients are now being admitted to the hospital. This may increase the number of patients in the normal BMI range without catabolic end-stage disease and account for similar mortality between patients without obesity and patients with Class I obesity. However, patients with Class I obesity did experience greater rates of ICU transfer with non-significant increased mortality. This may indicate underpowering in this subgroup and potentially increased mortality in Class I obesity compared to healthy patients with normal-range BMI.

While patients with obesity had worse clinical outcomes than those without obesity in our study, this effect does not appear to be mediated by a higher degree of inflammation. Markers of inflammation, including CRP and ferritin on the first and second hospital days, were lower among patients with obesity than those without. LDH was an exception to this general observation on the first hospital day. This increased LDH elevation in the 35–39.9 BMI group may reflect cell vulnerability and destruction rather than an IL-6 mediated acute inflammatory response. Increased expression of the inhibitory receptor NKG2A on CD8+ T-cells in Sars-CoV-2 infection has been correlated with a functional exhaustion of antiviral lymphocytes, priming them for destruction and in part causing the increased serum LDH [44]. The increased absolute number of adipocytes found centrally or ectopically in obesity [13] can lead to a sustained increase in circulating leptin, up-regulating Glut1 receptors on a variant of T-cell modulators. The subsequent shift to increased CD8+ T-cells [45] may result in more

vulnerable CD8+ T-cells, rapid cell turnover by direct destruction, and increased LDH over those without obesity.

Data on symptom duration were not available, and it is possible that patients with obesity presented to care earlier in the disease course due to reduced respiratory reserve and earlier onset of hypoxemia. This hypothesis is further supported by a trend for decreasing inflammatory markers with increasing BMI. Increasing BMI has been correlated with increasing A-a gradient, and other altered respiratory physiology which may predispose to hypoxemia and hypoventilation at baseline [7, 33, 34]. Patients without obesity may have been able to compensate for an increased arterial-alveolar gradient longer and therefore presented later. Indeed, previous studies have shown that individuals with obesity have increased ventilatory demand, increased work of breathing, decreased respiratory compliance, and respiratory muscle insufficiency [34] leading to difficulties with mechanical ventilation in the ICU. Further research is necessary to determine whether the change in respiratory reserve drives the increased need for ICU care and increased mortality.

Comorbidities associated with obesity may also play a role in worse outcomes in those with obesity. Patients with obesity are at higher risk of fatty liver disease, and greater viral invasion with organ dysfunction may contribute to the increased mortality seen in these patients. One study found that concurrent fatty liver disease and obesity increased risk of severe illness by 6-fold [46]. One possible mechanism is greater viral invasion of adipose tissue, as ACE2 expression has been found at high levels in adipose tissue [47].

Additional factors may explain the unexpected finding of lower inflammatory markers in patients with obesity. It is possible that individuals with obesity (and with chronic inflammation) mount a delayed inflammatory response after Covid-19—a phenomenon seen with influenza [48]. Previous studies have shown that individuals with obesity may have impaired macrophage activation and B- and T-cell responses after viral infection [14, 48] and their ability to control viral replication is impaired. Secondly, BMI may not be an adequate marker of visceral body fat; it serves as a mere surrogate for central or ectopic adiposity [13, 30, 37]. Hence, CRP levels may not have a direct relationship to BMI alone.

Notably, the inflammatory response varied when stratified by obesity and T2D. Those individuals with obesity and T2D had higher inflammatory markers on the first hospital day compared to those with obesity alone. This could reflect that those with T2D and obesity have more metabolic dysfunction and inflammation as opposed to those with obesity alone who may have been hospitalized with more anatomic dysfunction due to hypoventilation. T2D was not associated with increased ICU transfer or mortality nor was there any statistical interaction between T2D and obesity to alter clinical outcomes.

This study has numerous strengths. The study only included patients with PCR-confirmed SARS-CoV-2 infection, so we eliminated any misclassification of disease status. Further, these data are from a safety-net hospital with racial diversity and a high rate of comorbidities. Indeed, the rate of obesity in our population is higher than the US average: 45.6% vs 39.8% [49]. Not only was the size of our cohort large, but nearly all patients (99.7%) had outcome data at the time of data abstraction. As with any critical illness, patients with Covid-19 can at times survive with supportive care for several weeks leading to an underestimate of mortality if assessed too early in disease course.

As this is an observational study, causative conclusions cannot be drawn. Extraction of comorbidities and laboratory data is limited by appropriate chart documentation and laboratory orders by providers. Although we controlled for use of biologic therapy in our multivariate model, there were changes to the hospital treatment protocols between March and May that may have confounded our results. Further, although we assessed the impact of a T2D diagnosis on inflammatory markers and clinical outcomes, we did not quantify the degree of

dysglycemia using a marker such as the hemoglobin A1c. Additionally, we did not have access to accurate and consistent outpatient medication data. We did not assess trends in inflammatory markers on the day of intubation, which would be expected to be higher than those seen at presentation, but comparisons would be confounded by the use of immunomodulators. Additionally, the results only apply to hospitalized patients, and rates of hospitalization may vary among patients with and without obesity.

In conclusion, this retrospective cohort study of Covid-19 patients suggests increased risk of ICU transfer for patients with BMI >30 and increased risk of mortality for those with BMI>35; these outcomes do not appear to be mediated by an increased inflammatory response early in the hospital course. These outcomes may be due to higher risk of hypoxemia with baseline ventilation-perfusion mismatch, increased difficulty ventilating patients with obesity, or differential timing of acute-on-chronic inflammation. Studies are needed to determine whether this decreased inflammatory response persists during hospitalization, whether pro-inflammatory complications are seen less commonly among patients with obesity, and whether biologic therapy should be utilized differently in patients with obesity.

## Supporting information

**S1 Table. Median (range) inflammatory markers and absolute lymphocyte count on hospital days 1 and 2 as well as maximum FiO2 during hospitalization by body mass index (BMI) group.** CRP = C reactive protein, LDH = lactate dehydrogenase ALC = absolute lymphocyte count.
(PDF)

## Author Contributions

**Conceptualization:** Anahita Mostaghim, Pranay Sinha, Laura F. White, Caroline Apovian, Manish Sagar, Natasha S. Hochberg.

**Data curation:** Anahita Mostaghim, Catherine Bielick, Indeevar Beeram.

**Formal analysis:** Anahita Mostaghim, Pranay Sinha, Catherine Bielick, Selby Knudsen.

**Investigation:** Anahita Mostaghim.

**Methodology:** Laura F. White, Natasha S. Hochberg.

**Project administration:** Anahita Mostaghim.

**Supervision:** Natasha S. Hochberg.

**Writing – original draft:** Anahita Mostaghim.

**Writing – review & editing:** Anahita Mostaghim, Pranay Sinha, Catherine Bielick, Laura F. White, Caroline Apovian, Manish Sagar, Natasha S. Hochberg.

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
