## [Decision Letter · Decision Letter 0]

7 Nov 2020

PONE-D-20-30399

Clinical outcomes and inflammatory marker levels in patients with Covid-19 and obesity at an inner-city safety net hospital

PLOS ONE

Dear Dr. Hochberg,

Thank you for submitting your manuscript to PLOS ONE. After careful consideration, we feel that it has merit but does not fully meet PLOS ONE’s publication criteria as it currently stands. Therefore, we invite you to submit a revised version of the manuscript that addresses the points raised during the review process.

We look forward to receiving your revised manuscript.

Kind regards,

Aleksandar R. Zivkovic

Academic Editor

PLOS ONE

Journal Requirements:

2. In the ethics statement in the manuscript and in the online submission form, please provide additional information about the patient records used in your retrospective study, including:

a) whether all data were fully anonymized before you accessed them and/or whether the IRB or ethics committee waived the requirement for informed consent;

b) the date range (month and year) during which patients' medical records were accessed;

c) the source of the medical records analyzed in this work (e.g. hospital, institution or medical center name).

3. Please upload a new copy of Figure 1 as the detail is not clear. Please follow the link for more information: https://blogs.plos.org/plos/2019/06/looking-good-tips-for-creating-your-plos-figures-graphics/

Reviewers' comments:

Reviewer's Responses to Questions

**Comments to the Author**

Reviewer #1: It is important to put data also on diabetes patients with obesity, like glycemic control (HbA1c) and also medication before for DM, the same with hypertension what is the most drug they use regularly for hypertension

Reviewer #2: This is a retrospective study with 791 hospitalized patients with Covid-19 that was evaluated clinical outcomes and inflammatory marker levels according to obesity presence or not. The main results hightlight that patients with obesity had more chance to die but not necessarely had more inflammation. However, when they analysed the variables according to the presence of diabetes and obesity or obesity without diabetes these results change. We know that this is a retrospective study and as the authors discussed causative conclusions we could not state about these observations, but it is a very intriguing result. Maybe esteatohepatitis could be the cause for these differences. I think it would be very interesting if the authors could add any comments about this.

In general, it is a very interesting manuscript. Congratulations all the authors! Absolutely, it deserves to be published.

Reviewer #3: This topic is not novel, the association between obesity and adverse outcomes of COVID-19 has been established

However I recommend publication of the article, as it adds to the evidences about this topic

The discussion is not well developed and should include all papers that have been developed about this topic notably

https://www.ncbi.nlm.nih.gov/pmc/articles/PMC7314342/

https://www.ncbi.nlm.nih.gov/pmc/articles/PMC7513689/

6. PLOS authors have the option to publish the peer review history of their article (what does this mean?). If published, this will include your full peer review and any attached files.

Reviewer #1: No

Reviewer #2: **Yes: **Simone Cristina Soares Brandão

Reviewer #3: No

---

## [Author Response · Author response to Decision Letter 0]

24 Nov 2020

1. The first reviewer recommended “It is important to put data also on diabetes patients with obesity, like glycemic control (HbA1c) and also medication before for DM, the same with hypertension what is the most drug they use regularly for hypertension”. 

a. Unfortunately, due to the retrospective nature of the study, we were reliant on provider history-taking and orders. Many of our patients seek primary care outside our hospital system and do not remember medications on admission interview, so a complete and accurate picture of this is not able to be obtained. We have previously mentioned one of the limitations as not being able to obtain HgA1c measurements, however we have added the inability to obtain accurate pre-hospital medication lists to this. 

2. The second reviewer recommended “This is a retrospective study with 791 hospitalized patients with Covid-19 that was evaluated clinical outcomes and inflammatory marker levels according to obesity presence or not. The main results highlight that patients with obesity had more chance to die but not necessarily had more inflammation. However, when they analyzed the variables according to the presence of diabetes and obesity or obesity without diabetes these results change. We know that this is a retrospective study and as the authors discussed causative conclusions we could not state about these observations, but it is a very intriguing result. Maybe steatohepatitis could be the cause for these differences. I think it would be very interesting if the authors could add any comments about this.

In general, it is a very interesting manuscript. Congratulations all the authors! Absolutely, it deserves to be published.” 

a. We have added to the discussion section the potential mechanism of this including studies of steatohepatitis in Covid-19 and in sepsis.

3. The third reviewer recommended “This topic is not novel, the association between obesity and adverse outcomes of COVID-19 has been established. However I recommend publication of the article, as it adds to the evidences about this topic

The discussion is not well developed and should include all papers that have been developed about this topic notably” and listed two specific articles. 

a. One article brought up the relationship between higher obesity rates by country, GDP, and food supply with worse outcomes. Given the rapidly evolving nature of COVID and associated publications, many publications have come out. In order to update our paper from time of writing, we have included the recommended paper in our introduction as well as a newer meta-analysis that specifically addresses outcomes comparatively between patients with obesity and those without obesity in 35 published cohorts. We included a discussion with references on inflammatory mechanisms in obesity, which includes immune exhaustion, leptin cycling, inflammatory markers, and adipose tissue immune cells which already comprise the bulk of the review paper suggested. However, as our paper suggests a non-inflammatory component to increased mortality in COVID and obesity, we have cited a newer paper directly tying viral mechanism to adipose tissue.

---

## [Editor Report · Decision Letter 1]

1 Dec 2020

Clinical outcomes and inflammatory marker levels in patients with Covid-19 and obesity at an inner-city safety net hospital

PONE-D-20-30399R1

Dear Dr. Hochberg,

We’re pleased to inform you that your manuscript has been judged scientifically suitable for publication and will be formally accepted for publication once it meets all outstanding technical requirements.

Kind regards,

Aleksandar R. Zivkovic

Academic Editor

PLOS ONE
---

## [Editor Report · Acceptance letter]

11 Dec 2020

PONE-D-20-30399R1 

Clinical outcomes and inflammatory marker levels in patients with Covid-19 and obesity at an inner-city safety net hospital 

Dear Dr. Hochberg:

I'm pleased to inform you that your manuscript has been deemed suitable for publication in PLOS ONE. Congratulations! Your manuscript is now with our production department. 

Kind regards, 

on behalf of

Dr. Aleksandar R. Zivkovic 

Academic Editor

PLOS ONE